# Pooled testing of traced contacts under superspreading dynamics

**Stratis Tsirtsis**[1]*, **Abir De**[2], **Lars Lorch**[3], **Manuel Gomez-Rodriguez**[1]*

**1** Max Planck Institute for Software Systems, Kaiserslautern, Germany, **2** IIT Bombay, Mumbai, India, **3** ETH Zürich, Zürich, Switzerland

* stsirtsis@mpi-sws.org (ST); manuelgr@mpi-sws.org (MG-R)

## Abstract

Testing is recommended for all close contacts of confirmed COVID-19 patients. However, existing pooled testing methods are oblivious to the circumstances of contagion provided by contact tracing. Here, we build upon a well-known semi-adaptive pooled testing method, Dorfman's method with imperfect tests, and derive a simple pooled testing method based on dynamic programming that is specifically designed to use information provided by contact tracing. Experiments using a variety of reproduction numbers and dispersion levels, including those estimated in the context of the COVID-19 pandemic, show that the pools found using our method result in a significantly lower number of tests than those found using Dorfman's method. Our method provides the greatest competitive advantage when the number of contacts of an infected individual is small, or the distribution of secondary infections is highly overdispersed. Moreover, it maintains this competitive advantage under imperfect contact tracing and significant levels of dilution.

**Data Availability Statement:** Data and code necessary to replicate the results in this article are available at https://github.com/Networks-Learning/pooled-testing.

## Author summary

Due to the emergence of COVID-19, pooled testing has gained significant attention as a method for allocating testing resources more efficiently. In this context, the majority of existing pooled testing methods for the identification of infected individuals are agnostic to the circumstances of contagion. However, individuals for whom a test is ordered are usually traced contacts of an infectious person—they are secondary infections. As a result, their infection statuses are correlated. In this work, we propose a novel pooled testing method that makes explicit use of epidemic parameters describing the distribution of secondary infections. Our method partitions an infected individual's contacts into pools whose sizes make more efficient use of the available tests. Extensive simulations under a variety of epidemiological conditions informed by the COVID-19 literature show that our method can significantly decrease the expected number of tests under superspreading dynamics, i.e., when the distribution of secondary infections exhibits high variance. The simulations also show that our method maintains its advantageous performance under imperfect conditions, such as significant dilution effects or incomplete contact tracing.

**Funding:** This project has received funding (grant recipient: M.G.-R.) from the European Research Council (ERC) (https://erc.europa.eu/) under the European Union's Horizon 2020 research and innovation programme (grant agreement No. 945719). The funders had no role in study design, data collection and analysis, decision to publish, or preparation of the manuscript.

**Competing interests:** The authors have declared that no competing interests exist.

This is a *PLOS Computational Biology* Methods paper.

## Introduction

As countries around the world learn to live with COVID-19, the use of testing, contact tracing and isolation has been proven to be as important as social distancing for containing the spread of the disease [1,2]. However, as the infection levels grow, their effectiveness reaches a tipping point and quickly degrades since the health authorities lack resources to trace and test all contacts of a diagnosed individual [3]. In this context, there has been a flurry of interest in the use of pooled testing—testing pools of multiple samples simultaneously—to scale up testing under limited resources.

The literature on pooled testing methods has a rich history, starting with the seminal work by Dorfman [4]. However, the majority of existing methods [4–22], including those allowing for different individual infection probabilities [7,11–15] as well as those developed and used in the context of the COVID-19 pandemic [16–22], assume statistical *independence* of the samples to be tested. This assumption can be seemingly justified by classical epidemiological models where the number of infections caused by a single individual follows a Poisson distribution. However, for COVID-19, there is growing evidence suggesting that the number of secondary infections caused by a single individual is *overdispersed*—most individuals do not infect anyone but a few superspreaders infect many in infection hotspots [23–26] (Overdispersion has been also observed in MERS and SARS [27–30]). This suggests that the infection statuses of samples from close contacts of the same infected individual may be correlated. Only very recently, a narrow line of work has relaxed the above independence assumption [31–33]. However, these works only investigate to what extent the correlation between samples influences the expected number of tests in pooled testing, rather than proposing a method to find the optimal partition of correlated samples into pools. Furthermore, their investigations build upon infection probability distributions whose parameters may be difficult to estimate using real contact tracing data, reducing their potential applicability in practice.

In this work, we build upon a well-known semi-adaptive pooled testing method, Dorfman's method with imperfect tests [7,8,34]. In Dorfman's method, samples from multiple individuals are first pooled together and evaluated using a single test. If a pooled sample is negative, all individuals in the pooled sample are deemed negative. If the pooled sample is positive, each individual sample from the pool is then tested independently. To determine testing pools, Dorfman's method models the probability of individual samples being positive with independent and identically distributed (i.i.d.) Bernoulli distributions. Contrary to this, we assume that: (i) the samples to be tested are all the (close) contacts of a diagnosed individual during their infectious period who are identified using contact tracing, and (ii) the number of true positive samples, i.e., secondary infections by the diagnosed individual, follows an overdispersed generalized negative binomial distribution, as commonly done in epidemiological studies quantifying the superspreading of infectious diseases [23–25,28,30]. We introduce a dynamic programming algorithm to efficiently find a partition of the contacts into pools, possibly of different sizes, that optimally trade off the average number of tests, false negatives and false positives in polynomial time. Under our assumptions, contacts are exchangeable within pools, hence the optimal pools can be filled and tested sequentially as samples from contacts become available, as for Dorfman's method.

Experiments using a variety of reproduction numbers and dispersion levels in secondary infections, including those observed for COVID-19, show that the pools found using our method result in a significantly lower average number of tests than those found using the standard Dorfman's method. Our method provides the greatest competitive advantage when the number of contacts of an infected individual is small or the distribution of secondary infections is highly overdispersed. Moreover, it maintains this competitive advantage under imperfect contact tracing and significant levels of dilution.

## Methods

### Modeling overdispersion of infected contacts

Previous work has mostly built on the assumption that the number of infections X caused by a single individual follows a Poisson distribution with mean R, so X ~ Poisson(R), where R is often called the effective reproduction number. However, having equal mean and variance, the Poisson is unable to capture settings where the number of cases exhibits higher variance. Following recent work in the context of COVID-19 [24,25], we instead model X using a generalized negative binomial distribution. In a (standard) negative binomial distribution, X ~ NBin (k, p) can be interpreted as the number of successes before the k-th failure in a sequence of Bernoulli trials with success probability p. In a generalized negative binomial distribution, k > 0 can take real values and the probability mass function is given by

$$P(X = n) = \frac{\Gamma(n + k)}{\Gamma(k)n!} p^n (1 - p)^k,$$

where k is called the dispersion parameter and parameterizes higher variance of the distribution for small k. Here, we assume that the number of secondary infections X is distributed as X ~ NBin(k, p) with p = R / (k + R), hence parameterizing X via its mean $\mathbb{E}[X] = R$ and dispersion parameter k. Under this parameterization, Var[X] = R (1 + R /k), which is greater than the variance of the Poisson R for $k<\infty$. For $k\rightarrow\infty$, the sequence of random variables $X_k$ ~ NBin(k, R/ (k + R)) converges in distribution to X ~ Poisson(R).

Furthermore, since by assumption we identify all contacts of a diagnosed individual using contact tracing, we have prior information about the maximum number of possible infections N. More specifically, we can use the following truncated negative binomial distribution in our derivations:

$$q_{R,k,N}(n) := P(X = n | X \leq N), \tag{1}$$

where X ~ NBin(k, R/(k+R)) and note that P(X = n | X ≤ N) = P(X = n) / P(X ≤ N) if n ≤ N and 0 otherwise. In practice, the identification of all contacts of a diagnosed individual might not always be feasible, however, our method's competitive performance with respect to Dorfman's remains even if contact tracing is unable to identify all contacts of a diagnosed individual (refer to the Results section).

### Pooling contacts of a positively diagnosed individual

Our goal is to identify infected individuals among all contacts $\mathcal{N}$ of a positively diagnosed individual via testing, where $|\mathcal{N}| = N$. For each individual $j \in \mathcal{N}$, we define the indicator random variable $I_j = \mathbb{I}[\text{individual j is infected}]$ and, for each pool of individuals $\mathcal{S} \subseteq \mathcal{N}$, we define the number of infected in $\mathcal{S}$ as $I(\mathcal{S}) := \sum_{j \in \mathcal{S}} I_j$. Moreover, following our assumption on the distribution of the number of secondary infections, we define $P(I(\mathcal{N}) = n) = q_{R,k,N}(n)$.

Let $T(\mathcal{S}) = \mathbb{I}$[test of pool $\mathcal{S}$ is positive]. To account for imperfect tests, we specify the sensitivity $s_e$ (i.e., true positive probability) and the specificity $s_p$ (i.e., true negative probability) of individual testing. To capture the effect of dilution when testing a pool $\mathcal{S}$, we adopt the model of Burns and Mauro [35] and parameterize the conditional probabilities as

$$P(T(\mathcal{S}) = 1 | I(\mathcal{S}) = s > 0) = 1 - s_p + \left(s_e + s_p - 1\right)\left(\frac{s}{|\mathcal{S}|}\right)^d$$

$$P(T(\mathcal{S}) = 0 | I(\mathcal{S}) = 0) = s_p.$$

Here, $d \in [0, 1]$ controls the effect that dilution has on a pooled test's sensitivity. The right-hand side of the top equation converges to $s_e$ as $d \to 0$. In the above, we implicitly assume that all infected individuals contribute equally to the concentration of viral load in a pool and the probability of a false positive pooled test is independent of the size of the pool since the concentration of the virus is zero in any case.

## Dorfman testing under overdispersion of infected contacts

Dorfman testing proceeds by pooling individuals into non-overlapping partitions of $\mathcal{N}$ and first testing the combined samples of each pool using a single test. Every member of a pool is marked as negative if their combined sample is negative. In contrast, if a combined sample of a pool is positive, each individual of the pool is subsequently tested individually to determine who exactly is marked positive in the pool.

Let $D_j^{\mathcal{S}}$ denote the indicator random variable for the event that individual j is marked as infected in pool $\mathcal{S} \subseteq \mathcal{N} : |\mathcal{S}| > 1$ after Dorfman testing. Then, its value can be expressed as

$$D_j^{\mathcal{S}} = \mathbb{I}[T(\mathcal{S}) = 1 \cap T(\{j\}) = 1],$$

i.e., it takes the value 1 if and only if the combined sample of pool $\mathcal{S}$ is first tested positive and subsequently the sample of individual j is tested positive. In the simple case of $|\mathcal{S}| = 1$, we have $D_j^{\mathcal{S}} = T(\{j\})$.

## Finding the optimal pool sizes

We first compute the expected number of tests, false negatives, and false positives due to each pool $\mathcal{S}$ for Dorfman testing under our above model of infected contacts. Their values only depend on the pool size (refer to S1 Appendix). Hence, for a given number of contacts $|\mathcal{N}| = N$ and pool of size $|\mathcal{S}| = s$, we overload notation and write $\mathbb{E}[K(s)]$, $\mathbb{E}[FN(s)]$ and $\mathbb{E}[FP(s)]$ for the expected number of tests, false negatives and false positives, respectively.

Let $\mathcal{Z}$ be the set of all sets $\mathcal{C}$ of positive integers such that $1 \leq |\mathcal{C}| \leq N$ and $\sum_{s \in \mathcal{C}} s = N$. It is easy to see that every such set $\mathcal{C} = \{s_1, s_2, \ldots, s_C\}$ is a valid partition of the set of contacts $\mathcal{N}$ into a set of pools with sizes $s_1, s_2, \ldots, s_C$. In that context, our goal is to find the sizes $\mathcal{C}$ of the sets of pools that optimally trade off the expected number of tests, false negatives and false positives [8,35]:

$$\underset{\mathcal{C} \in \mathcal{Z}}{\text{minimize}} \sum_{s \in \mathcal{C}} g(s),$$

with $g(s) = \mathbb{E}[K(s)] + \lambda_1 \mathbb{E}[FN(s)] + \lambda_2 \mathbb{E}[FP(s)]$, where $\lambda_1$ and $\lambda_2$ are given nonnegative parameters that balance the penalty incurred by false negatives and false positives. Note that the parameters $\lambda_1, \lambda_2$ can be thought of as Lagrange multipliers for the problem of minimizing the expected number of tests subject to the expected numbers of false negatives and false

positives being less than two given values. For a discussion on alternative objective functions and their benefits, we refer the interested reader to [36,37].

Perhaps surprisingly, we can solve the above problem in polynomial time using a simple dynamic programming procedure. To do so, we define the following recursive functions:

$$h(n) = \min_{1 \le j \le n}[g(j) + h(n - j)] \text{ and } \mathcal{S}_n = \mathcal{S}_{n-s} \cup \{s\},$$

where $s = \text{argmin}_{1 \le j \le n}[g(j) + h(n - j)]$. Interpreting n as the number of individuals not yet assigned to a pool, using the two recursive functions, the (sizes of the) optimal sets of pools can be recovered by computing h(n) in increasing order of n, up to the value N. Refer to S2 Appendix for pseudocode summarizing the overall procedure and a formal proof of optimality. If the testing authorities wish to manually assign a given fraction of x contacts to pools based on some other criteria (e.g., household membership [38]), the optimal sets of pools for the remaining N-x contacts can be recovered by computing h(n) in increasing order of n, up to the value N-x.

## Experimental design

We perform simulations to compare our method against Dorfman's method in terms of its ability to optimally trade off resources and false test outcomes in the presence of overdispersed distributions of secondary infections. Although it is possible to derive analytical expressions for each method's expected numbers of tests, false negatives and false positives, we resort to simulations to fully characterize and compare their (empirical) distributions. To evaluate the performance of the two methods, we generate the infection states of a set of contacts by first fixing the number of contacts N and sampling the secondary infections $n \sim q_{R,k,N}(n)$, where $q_{R,k,N}(n)$ is a truncated negative binomial distribution as defined in Eq 1. Then, we select n of the N contacts at random and set their status to be infected. To find the optimal pool sizes given by our method, we use our dynamic programming algorithm and the expected numbers of tests, false negatives and false positives, computed assuming the same truncated negative binomial distribution of secondary infections (refer to S1 Appendix). To find the optimal pool sizes given by Dorfman's method, we use a variation of the dynamic programming algorithm in which the expected numbers of tests, false negatives and false positives are computed assuming an i.i.d. probability of infection of $p = \mathbb{E}_{q_{R,k,N}}[n]/N$ for each individual contact (refer to S3 Appendix).

Following the literature on COVID-19, we consider (PCR) tests with high specificity and moderate sensitivity [39]. It is worth noting that we distinguish between two types of sensitivity and specificity: analytic and clinical. The former is reflecting a test's accuracy in a controlled laboratory environment while the latter is also affected by factors related to sample collection (e.g., stage of the disease at the time of collection, use of a throat or nasal swab) and, therefore, it is typically lower [40]. Since we focus on pooled testing of samples obtained through contact tracing, we will generally refer to clinical sensitivity and specificity unless otherwise specified. In this context, most studies report values in the range of 70% - 98% and 97% - 99% for individual tests' (clinical) sensitivity and specificity, respectively, with the exact value differing based on the method of sample collection and the laboratory protocol followed [41–43]. Informed by these values, we set $s_e = 0.8$ and $s_p = 0.98$. However, we provide additional results for alternative $s_e$, $s_p$ values in the Supporting information section.

To set the value of the parameter d that controls the effect of dilution, we fit the parameterized expression of the conditional probability $P(T(\mathcal{S}) = 1 | I(\mathcal{S}) = s > 0)$ to real pooled testing data analyzed by Bateman et al. [44]. In this study, the authors report that a PCR test's analytic

sensitivity for an undiluted sample equals to 0.99, whereas this sensitivity drops to 0.93, 0.91 and 0.81 when a single infected sample is present in pools of 5, 10 and 50 respectively. The analytic specificity of the test is not reported and hence, we assume that it is also equal to 0.99. Using these values, we get an estimate of d = 0.0455 via ridge regression, which we use throughout the paper. The resulting curve showing the effect of dilution on the sensitivity of a pooled test as a function of the virus's concentration is depicted in S1 Fig.

## Results

### Reduction in the average number of tests compared to Dorfman's method

We first compare the performance of our method and Dorfman's method in finding the pools that minimize the average number of tests (i.e., $\lambda_1 = \lambda_2 = 0$) for fixed values of the reproductive number R and the dispersion parameter k, matching estimates obtained during the early phase of the COVID-19 pandemic [24]. Table 1 summarizes the results for different numbers of contacts N of the diagnosed individuals. The results show that our method achieves a lower average number of tests across all settings and indicate greater competitive advantage when the number of contacts is small. These results hold across a variety of sensitivity and specificity values (refer to S1 and S2 Tables and S2 Fig). At the same time, the average numbers of false negatives and false positives are similar for the two methods. That said, we observe that both methods present high variance, with ours' being generally larger. Looking at the average pool sizes chosen by each method in Fig 1A, we observe that Dorfman's method chooses smaller pool sizes that increase with the number of contacts while the ones chosen by our method remain relatively constant. This leads to significant differences between the distributions of the number of tests performed under the two methods. For example, as shown in Fig 1B, when the number of contacts is N = 20, our method is most likely to perform about 70% less tests than Dorfman's. However, due to the more conservative pool sizes selected by Dorfman's method, there is a small probability that our method ends up performing more tests, sometimes even double the amount.

Next, we investigate to what extent our method improves upon Dorfman's method for other values of the reproductive number R and dispersion parameter k, including those estimated by several COVID-19 studies [23–25,45–48]. Fig 2 summarizes the results, which show that our method offers the greatest competitive advantage whenever the reproductive number R is large and the number of secondary infections is overdispersed, i.e., k → 0. The results suggest that for an infected individual with N = 100 contacts and under the estimated values of reproductive number and dispersion parameter reported in the COVID-19 literature, our method would have saved 3%-30% with respect to Dorfman's method. Similar findings hold for a variety of values for the number of contacts N, sensitivity $s_e$ and specificity $s_p$ (refer to S3–S5 Figs).

**Table 1. Average numbers of tests, false negatives and false positives achieved by our method (Dorf-OD) and classic Dorfman's method (Dorf-Cl) for various values of the number of contacts N.** Here, we sample the number of secondary infections from a truncated negative binomial distribution with reproductive number R = 2.5 and dispersion parameter k = 0.1 [24] and, we set the sensitivity and specificity to $s_e$ = 0.8, $s_p$ = 0.98. For each combination of method and parameter values, the averages and standard deviations are estimated using 10,000 samples.

| N | Average # of tests per contact | | Average # of false negatives per contact | | Average # of false positives per contact | |
|---|---|---|---|---|---|---|
| | Dorf-Cl | Dorf-OD | Dorf-Cl | Dorf-OD | Dorf-Cl | Dorf-OD |
| 20 | 0.331 (σ: 0.250) | 0.245 (σ: 0.396) | 0.024 (σ: 0.070) | 0.025 (σ: 0.087) | 0.002 (σ: 0.009) | 0.003 (σ: 0.013) |
| 50 | 0.259 (σ: 0.220) | 0.219 (σ: 0.324) | 0.016 (σ: 0.050) | 0.016 (σ: 0.056) | 0.002 (σ: 0.007) | 0.003 (σ: 0.009) |
| 100 | 0.207 (σ: 0.184) | 0.180 (σ: 0.239) | 0.009 (σ: 0.030) | 0.009 (σ: 0.032) | 0.002 (σ: 0.005) | 0.002 (σ: 0.006) |
| 200 | 0.164 (σ: 0.149) | 0.148 (σ: 0.201) | 0.005 (σ: 0.016) | 0.005 (σ: 0.016) | 0.001 (σ: 0.004) | 0.002 (σ: 0.005) |

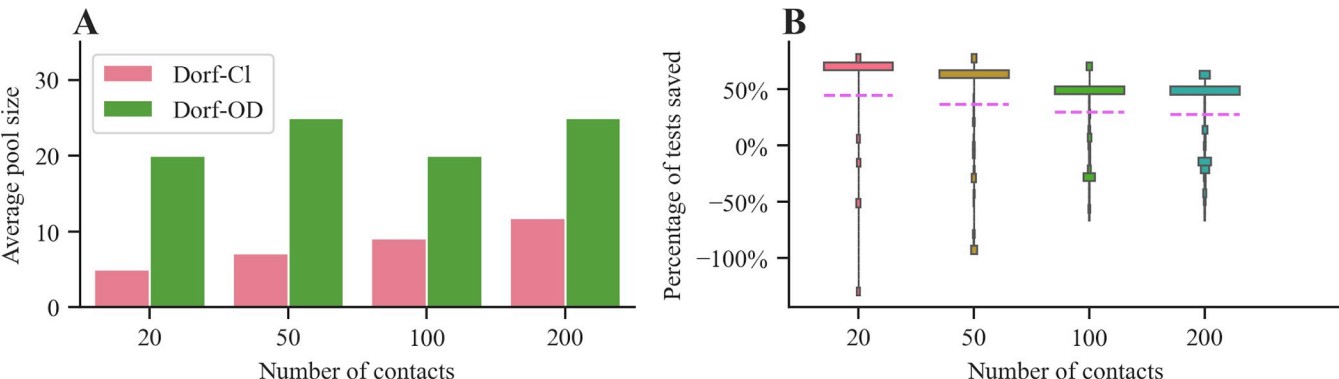

**Fig 1. Performance of our method (Dorf-OD) and classic Dorfman's method (Dorf-Cl) for various values of the number of contacts N of a diagnosed individual during their infectious period.** Panel (**A**) shows the average pool size. Panel (**B**) shows the empirical distribution of the percentage of tests saved by using our method instead of Dorfman's method, where we exclude the highest and lowest 5% of observations and the purple dashed lines represent average values. In both panels, we sample the number of secondary infections from a truncated negative binomial distribution with reproductive number R = 2.5 and dispersion parameter k = 0.1 [24] and, we set the sensitivity and specificity to $s_e = 0.8$, $s_p = 0.98$. For each combination of method and parameter values, the averages and quantiles in both panels are estimated using 10,000 samples.

## Balancing tests, false negatives and false positives

To explore the trade-off between the average number of tests that our method achieves and the false positive and negative rates, we experiment with different values of the parameters $\lambda_1$, $\lambda_2$ and the sensitivity $s_e$ and specificity $s_p$. Fig 3 summarizes the results, which show that to achieve lower false negative and false positive rates, more tests need to be performed. When

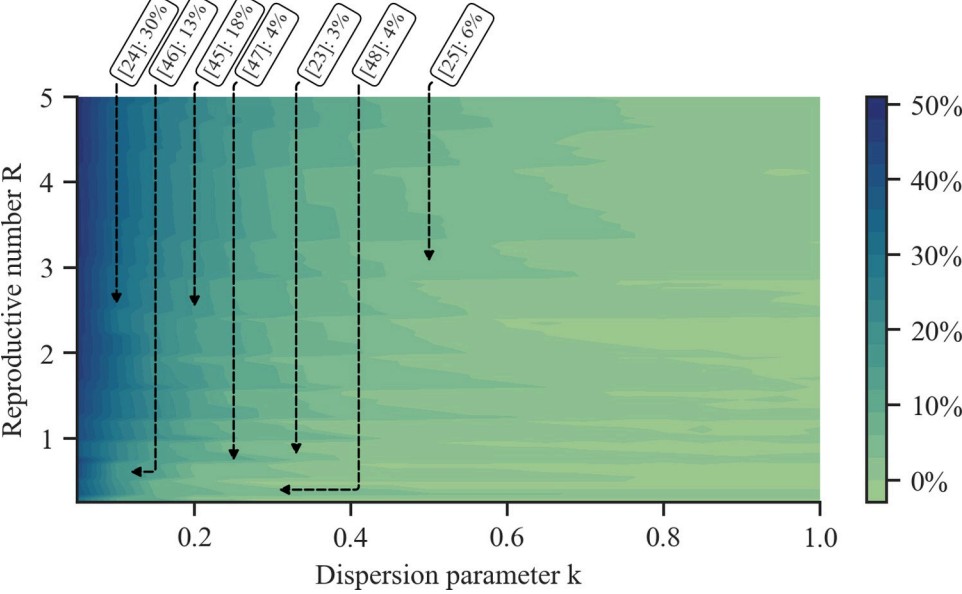

**Fig 2. Percentage of tests saved by using our method instead of Dorfman's method for different values of the reproductive number R and dispersion parameter k.** Darker colors correspond to a higher average percentage of tests saved. To generate the contour, we evaluate the average percentage of tests saved using values in [0.25, 5.0] with step 0.05 for R and in [0.05, 1.0] with step 0.05 for k. The overlaid annotations indicate the average percentage of tests saved for several estimated values of the reproductive number and dispersion parameter reported in the COVID-19 literature [23–25,45–48]. Here, we set the number of contacts to N = 100 and the sensitivity and specificity to $s_e = 0.8$, $s_p = 0.98$. In each experiment we estimate the average using 10,000 samples.

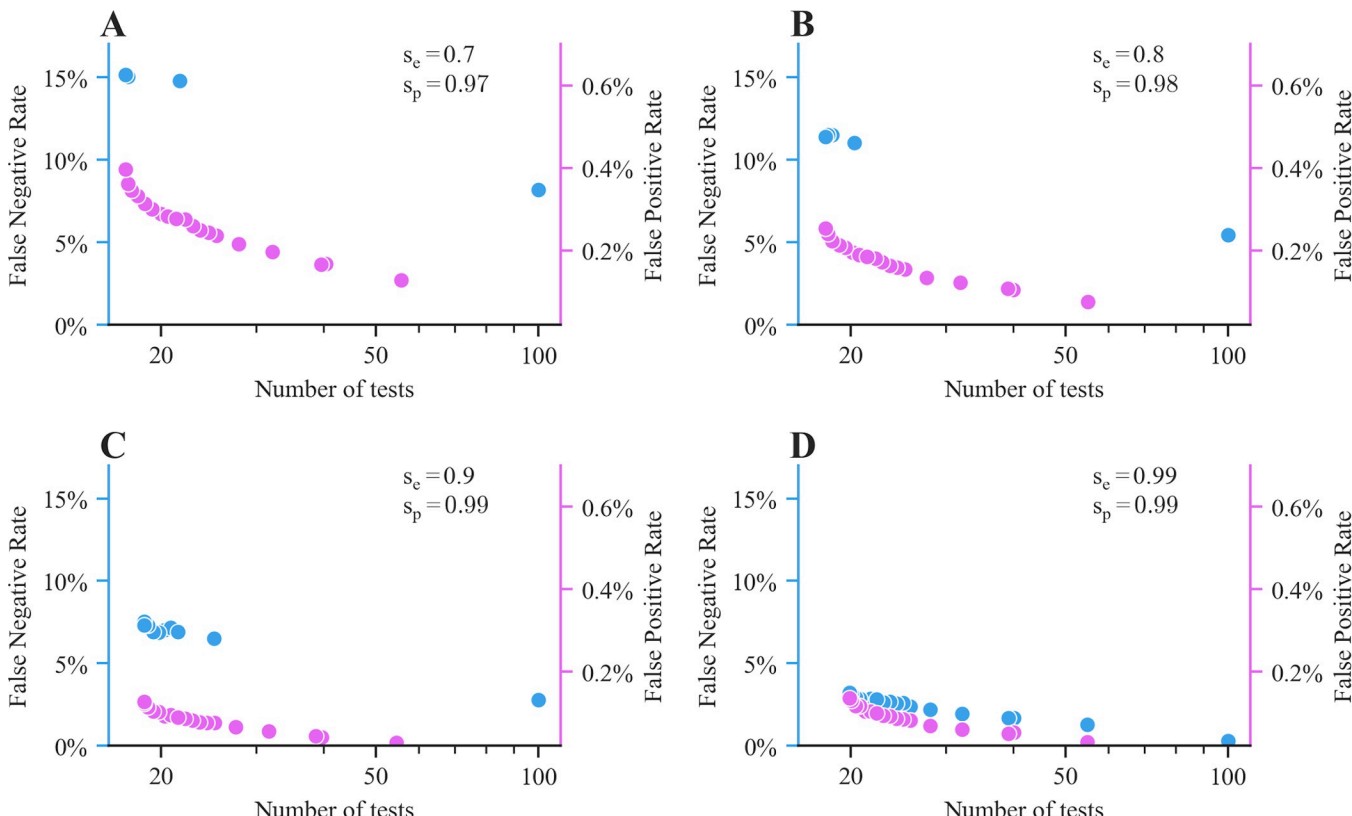

**Fig 3. Average number of tests, false negative rate and false positive rate achieved by our method under different values of the parameters $\lambda_1$ and $\lambda_2$ and different levels of specificity $s_e$ and sensitivity $s_p$.** In each panel, we either penalize the false negative rate (i.e., we vary $\lambda_1$ and set $\lambda_2 = 0$) or the false positive rate (i.e., we vary $\lambda_2$ and set $\lambda_1 = 0$). Accordingly, for the former, we show the false negative rate vs average number of tests (in blue) and, for the latter, we show the false positive rate vs average number of tests (in purple). Here, we set the number of contacts to N = 100 and sample the number of positive infections from a truncated negative binomial distribution with reproductive number R = 2.5 and dispersion parameter k = 0.1. In each experiment, we estimate averages using 10,000 samples. For the exact sizes of the optimal pools corresponding to each point in the figure, refer to S4–S11 Tables.

trading off the number of tests with the number of false positives ($\lambda_1 = 0$, $\lambda_2 > 0$), our method gradually changes the average pool size, leading to many unique pool partitions across $\lambda_2$ values. For small values of $\lambda_2$, the optimal solution leads to pool sizes that mainly minimize the number of tests. For large values of $\lambda_2$, the optimal solution consists of pools of two contacts. When balancing the number of tests with the number of false negatives ($\lambda_1 > 0$, $\lambda_2 = 0$) under the most realistic values of sensitivity and specificity, we observe that our method results in a small number of unique pool partitions across $\lambda_1$ values. For small values of $\lambda_1$, the optimal solution leads to pool sizes that mainly minimize the number of tests, similarly as in the previous case. For large values of $\lambda_1$, our method reaches a tipping point, after which, the optimal solution corresponds to individual testing (i.e., pools of size one). In contrast, when both the sensitivity and specificity are high ($s_e = s_p = 0.99$), we notice that the number of unique pool partitions increases. This indicates that when testing authorities have low tolerance for false negatives in the presence of significantly imperfect tests (i.e., when the value of $\lambda_1$ is large), reducing the pool size contributes marginally towards the reduction of false negative outcomes and individual testing becomes necessary. For the exact partitions into pools given by our method as we vary the values of $\lambda_1$ and $\lambda_2$, refer to S4–S11 Tables. Finally, note that, for large values of $\lambda_1$ ($\lambda_2$), Dorfman's method also results in pools of size one (two) and, therefore, the two methods become equivalent.

### On the effect of dilution

To assess the effect of dilution on the performance of our method and Dorfman's method at minimizing the average number of tests (i.e., $\lambda_1 = \lambda_2 = 0$), we experiment with different values of the parameter d, which controls the effect of dilution on the sensitivity of a pooled test. Fig 4 summarizes the results. As expected, Fig 4A shows that the average number of tests (false negatives) decreases (increases) as the level of dilution increases. However, we observe that our method presents a clear advantage when d < 0.6, which it achieves by favoring larger group sizes, as shown in Fig 4B. Our method never performs worse than Dorfman's method across the entire range of dilution levels. In this context, we point out that realistic values of the parameter d might lie in the lower range of the spectrum—our estimate about the dilution parameter based on data by Bateman et al. [44] gives d = 0.0455 while other studies in the context of COVID-19 report even weaker dilution effects (e.g., Yelin et al. [49] report an analytic sensitivity of 96% for pools of size 10). Therefore, we can conclude that our method would achieve a competitive advantage over Dorfman's method even if the dilution parameter d was slightly misspecified.

### Performance in the presence of unreported contacts

So far, we assumed that all close contacts of an infected individual are identified via contact tracing. Here, we study to what extent our method would be favorable over Dorfman's if contact tracing is incomplete, i.e., the true number of close contacts of an infected individual is underreported. As previously, we sample the infection statuses for an individual with a set of contacts $\mathcal{N}_{total}$, but we assume that only a random subset $\mathcal{N}_{traced}$ of fixed size $N = |\mathcal{N}_{traced}|$ is reported and tested. Fig 5 summarizes the results, which show that our method maintains its advantage at saving tests in comparison to Dorfman's method even when half of the individual's contacts are not reported to the contact tracing authorities. We also observe that the average percentage of tests saved by our method compared to Dorfman's increases as the effectiveness of contact tracing declines and the number of infected individuals among the set of traced contacts becomes smaller.

### Discussion

We have introduced a pooled testing method based on Dorfman's method that is especially designed to use information provided by contact tracing. In comparison with Dorfman's

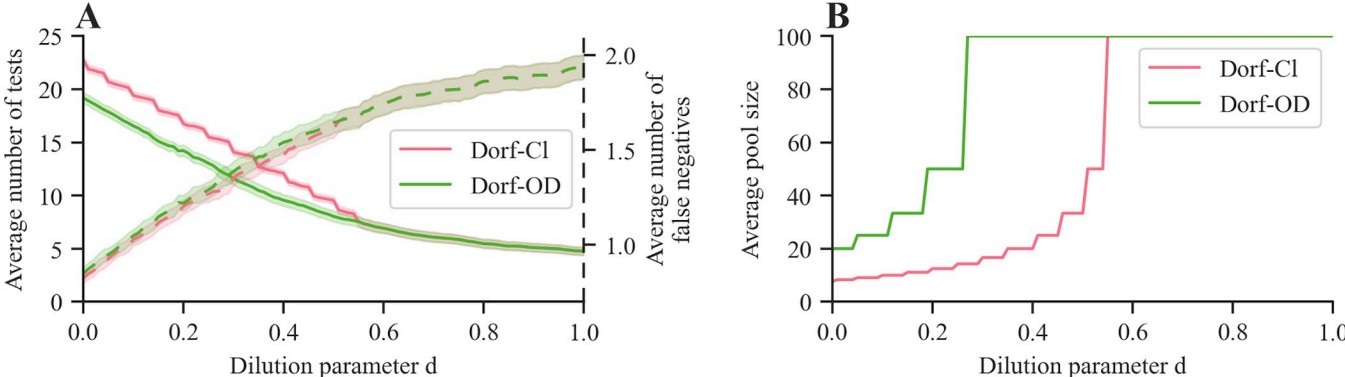

**Fig 4. Performance of our method (Dorf-OD) and classic Dorfman's method (Dorf-Cl) for various values of the dilution parameter d.** Panel (**A**) shows the average numbers of tests (solid lines) and false negatives (dashed lines). Panel (**B**) shows the average pool size. In both panels, shaded regions represent 95% confidence intervals. Here, we set N = 100, R = 2.5, k = 0.1, $s_e$ = 0.8, $s_p$ = 0.98 and, for each combination of method and parameter value, the averages are estimated using 10,000 samples.

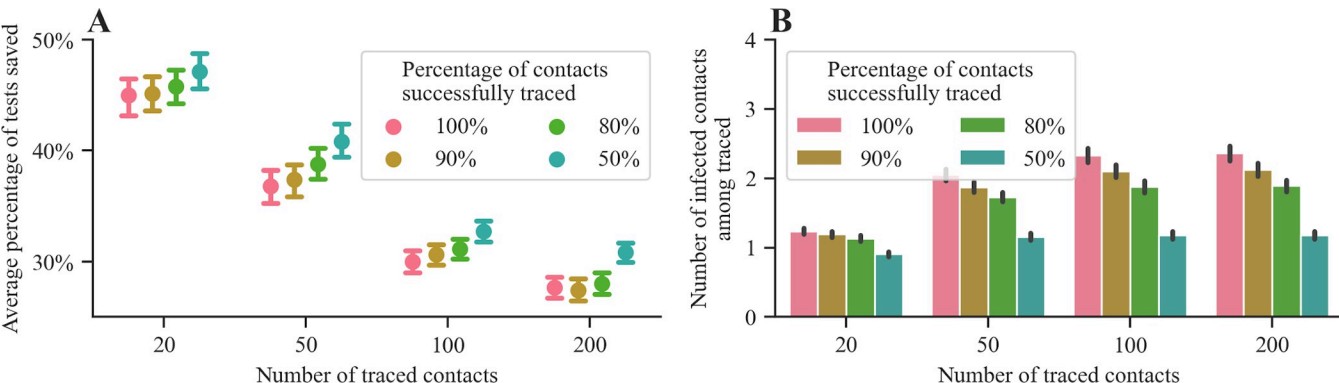

**Fig 5. Performance of our method and Dorfman's method under incomplete contact tracing.** Panel (**A**) shows the average percentage of tests saved by using our method instead of Dorfman's under various values of the number of traced contacts $N = |\mathcal{N}_{traced}|$ and the percentage of contacts who were successfully traced, i.e., $N/|\mathcal{N}_{total}|$. Panel (**B**) shows the number of infected contacts in $\mathcal{N}_{traced}$. In both panels, error bars represent 95% confidence intervals. Here, we first sample the number of positive infections from a truncated negative binomial distribution with reproductive number R = 2.5 and dispersion parameter k = 0.1 for a set of contacts $\mathcal{N}_{total}$ and then, we compute pool sizes and evaluate both methods based on a random subset of size N. For each combination of method and parameter values, the averages in all panels are estimated using 10,000 samples.

method, we showed through realistic simulations that our method finds pools that lead to a significant reduction in tests performed under a variety of epidemiological conditions, including those observed for the COVID-19 pandemic. Moreover, we also demonstrated that our method maintains its competitive advantage with respect to Dorfman's method under imperfect contact tracing and significant levels of dilution.

Our results have direct implications for the allocation of limited and imperfect testing resources in future pandemics whenever there exists evidence of substantial overdispersion in the number of secondary infections. However, we acknowledge that more research is needed to more accurately characterize the level of overdispersion in a pandemic, which is a prerequisite for our method to operate. In this context, it would be interesting to extend our approach using distributions other than the generalized negative binomial, which might reflect the number of secondary infections more suitably in different contact tracing scenarios. Moreover, it would be worth exploring alternative dilution models and objective functions. Another limiting factor of our method, which however holds for many pooled testing methods, is the assumption that the algorithm deciding about the partition of contacts into pools has access to the true sensitivity and specificity, which may not be trivial in practice [50]. A potential avenue for future work would be to investigate the impact of different testing methods (including ours) on the evolution of an epidemic under a limited testing capacity, using individual-based models [51–53]. Finally, to make our method applicable and beneficial for real contact tracing and pooled testing operations, it would be interesting to validate its reduced consumption of tests with respect to Dorfman's in randomized control studies.

## Supporting information

**S1 Fig. Effect of dilution on a pooled test's sensitivity.** The two lines show the sensitivity of a pooled test as a function of the concentration of viral load based on the parameterized model of $P(T(\mathcal{S}) = 1|I(\mathcal{S}) = s > 0)$. The green line shows a pooled test's analytic sensitivity (high $s_e$, $s_p$ values) which is fitted based on dilution data by Bateman et al. [44] and gives an estimate of d = 0.0455, via ridge regression. The blue line shows a pooled test's clinical sensitivity (moderate $s_e$ and high $s_p$ values) under the same value of the dilution parameter d.
(TIF)

**S2 Fig. Performance of our method (Dorf-OD) and classic Dorfman's method (Dorf-Cl) for various values of the number of contacts N, under additional levels of sensitivity $s_e$ and specificity $s_p$.** In panels (**A, B**) we set $s_e = 0.7$, $s_p = 0.97$, in panels (**C, D**) we set $s_e = 0.9$, $s_p = 0.99$ and, in panels (**E, F**) we set $s_e = 0.99$, $s_p = 0.99$. Panels (**A, C, E**) show the average pool size. Panels (**B, D, F**) show the empirical distribution of the percentage of tests saved by using our method instead of Dorfman's method, where we exclude the highest and lowest 5% of observations and the purple dashed lines represent average values. In all panels, we sample the number of secondary infections from a truncated negative binomial distribution with reproductive number R = 2.5 and dispersion parameter k = 0.1 [24]. For each combination of method and parameter values, the averages and quantiles in all panels are estimated using 10,000 samples.
(TIF)

**S3 Fig. Percentage of tests saved by using our method instead of Dorfman's method for different values of the reproductive number R and dispersion parameter k, under additional levels of sensitivity $s_e$, specificity $s_p$ and numbers of contacts N.** In panels (**A, B**), we set **N = 20** and **N = 50** respectively and, in both panels, we set the sensitivity and specificity to $s_e = 0.8$, $s_p = 0.98$. Darker colors correspond to a higher average percentage of tests saved. To generate the contours, we evaluate the average percentage of tests saved using values in [0.25, 5.0] with step 0.05 for R and in [0.05, 1.0] with step 0.05 for k. The overlaid annotations indicate the average percentage of tests saved for several estimated values of the reproductive number and dispersion parameter reported in the COVID-19 literature [23–25,45–48]. In each experiment, we estimate the average using 10,000 samples.
(TIF)

**S4 Fig. Percentage of tests saved by using our method instead of Dorfman's method for different values of the reproductive number R and dispersion parameter k, under additional levels of sensitivity $s_e$, specificity $s_p$ and numbers of contacts N.** In panels (**A, B, C**), we set **N = 20**, **N = 50** and **N = 100** respectively and, in all panels, we set the sensitivity and specificity to $s_e = 0.7$, $s_p = 0.97$. Darker colors correspond to a higher average percentage of tests saved. To generate the contours, we evaluate the average percentage of tests saved using values in [0.25, 5.0] with step 0.05 for R and in [0.05, 1.0] with step 0.05 for k. The overlaid annotations indicate the average percentage of tests saved for several estimated values of the reproductive number and dispersion parameter reported in the COVID-19 literature [23–25,45–48]. In each experiment, we estimate the average using 10,000 samples.
(TIF)

**S5 Fig. Percentage of tests saved by using our method instead of Dorfman's method for different values of the reproductive number R and dispersion parameter k, under additional levels of sensitivity $s_e$, specificity $s_p$ and numbers of contacts N.** In panels (**A, B, C**), we set **N = 20**, **N = 50** and **N = 100** respectively and, in all panels, we set the sensitivity and specificity to $s_e = 0.9$, $s_p = 0.99$. Darker colors correspond to a higher average percentage of tests saved. To generate the contours, we evaluate the average percentage of tests saved using values in [0.25, 5.0] with step 0.05 for R and in [0.05, 1.0] with step 0.05 for k. The overlaid annotations indicate the average percentage of tests saved for several estimated values of the reproductive number and dispersion parameter reported in the COVID-19 literature [23–25,45–48]. In each experiment, we estimate the average using 10,000 samples.
(TIF)

**S1 Table. Average numbers of tests, false negatives and false positives of our method (Dorf-OD) and classic Dorfman's method (Dorf-Cl) for various values of the number of**

contacts N, under additional levels of sensitivity $s_e$ and specificity $s_p$. Here, we set the sensitivity and specificity to $s_e = 0.7$, $s_p = 0.97$. We sample the number of secondary infections from a truncated negative binomial distribution with reproductive number R = 2.5 and dispersion parameter k = 0.1 [24] and, for each combination of method and parameter values, the averages and standard deviations are estimated using 10,000 samples.
(DOCX)

**S2 Table. Average numbers of tests, false negatives and false positives of our method (Dorf-OD) and classic Dorfman's method (Dorf-Cl) for various values of the number of contacts N, under additional levels of sensitivity $s_e$ and specificity $s_p$.** Here, we set the sensitivity and specificity to $s_e = 0.9$, $s_p = 0.99$. We sample the number of secondary infections from a truncated negative binomial distribution with reproductive number R = 2.5 and dispersion parameter k = 0.1 [24] and, for each combination of method and parameter values, the averages and standard deviations are estimated using 10,000 samples.
(DOCX)

**S3 Table. Average numbers of tests, false negatives and false positives of our method (Dorf-OD) and classic Dorfman's method (Dorf-Cl) for various values of the number of contacts N, under additional levels of sensitivity $s_e$ and specificity $s_p$.** Here, we set the sensitivity and specificity to $s_e = 0.99$, $s_p = 0.99$. We sample the number of secondary infections from a truncated negative binomial distribution with reproductive number R = 2.5 and dispersion parameter k = 0.1 [24] and, for each combination of method and parameter values, the averages and standard deviations are estimated using 10,000 samples.
(DOCX)

**S4 Table. Pool partitions corresponding to the points of Fig 3A, resulting by penalizing the false negative rate.** Here, under $s_e = 0.7$, $s_p = 0.97$, we **vary $\lambda_1$** while we fix $\lambda_2 = 0$ and, for each resulting partition, we compute the average number of tests and false negative/positive rate. We set the number of contacts to N = 100 and sample the number of positive infections from a truncated negative binomial distribution with reproductive number R = 2.5 and dispersion parameter k = 0.1. In each experiment, we estimate averages using 10,000 samples. Double entries in the first column correspond to cases where the set of contacts is partitioned into a combination of pools of two different sizes.
(DOCX)

**S5 Table. Pool partitions corresponding to the points of Fig 3A, resulting by penalizing the false positive rate.** Here, under $s_e = 0.7$, $s_p = 0.97$, we **vary $\lambda_2$** while we fix $\lambda_1 = 0$ and, for each resulting partition, we compute the average number of tests and false negative/positive rate. We set the number of contacts to N = 100 and sample the number of positive infections from a truncated negative binomial distribution with reproductive number R = 2.5 and dispersion parameter k = 0.1. In each experiment, we estimate averages using 10,000 samples. Double entries in the first column correspond to cases where the set of contacts is partitioned into a combination of pools of two different sizes.
(DOCX)

**S6 Table. Pool partitions corresponding to the points of Fig 3B, resulting by penalizing the false negative rate.** Here, under $s_e = 0.8$, $s_p = 0.98$, we **vary $\lambda_1$** while we fix $\lambda_2 = 0$ and, for each resulting partition, we compute the average number of tests and false negative/positive rate. We set the number of contacts to N = 100 and sample the number of positive infections from a truncated negative binomial distribution with reproductive number R = 2.5 and dispersion parameter k = 0.1. In each experiment, we estimate averages using 10,000 samples. Double

entries in the first column correspond to cases where the set of contacts is partitioned into a combination of pools of two different sizes.
(DOCX)

**S7 Table. Pool partitions corresponding to the points of Fig 3B, resulting by penalizing the false positive rate.** Here, under $s_e = 0.8$, $s_p = 0.98$, we **vary** $\lambda_2$ while we fix $\lambda_1 = 0$ and, for each resulting partition, we compute the average number of tests and false negative/positive rate. We set the number of contacts to N = 100 and sample the number of positive infections from a truncated negative binomial distribution with reproductive number R = 2.5 and dispersion parameter k = 0.1. In each experiment, we estimate averages using 10,000 samples. Double entries in the first column correspond to cases where the set of contacts is partitioned into a combination of pools of two different sizes.
(DOCX)

**S8 Table. Pool partitions corresponding to the points of Fig 3C, resulting by penalizing the false negative rate.** Here, under $s_e = 0.9$, $s_p = 0.99$, we **vary** $\lambda_1$ while we fix $\lambda_2 = 0$ and, for each resulting partition, we compute the average number of tests and false negative/positive rate. We set the number of contacts to N = 100 and sample the number of positive infections from a truncated negative binomial distribution with reproductive number R = 2.5 and dispersion parameter k = 0.1. In each experiment, we estimate averages using 10,000 samples. Double entries in the first column correspond to cases where the set of contacts is partitioned into a combination of pools of two different sizes.
(DOCX)

**S9 Table. Pool partitions corresponding to the points of Fig 3C, resulting by penalizing the false positive rate.** Here, under $s_e = 0.9$, $s_p = 0.99$, we **vary** $\lambda_2$ while we fix $\lambda_1 = 0$ and, for each resulting partition, we compute the average number of tests and false negative/positive rate. We set the number of contacts to N = 100 and sample the number of positive infections from a truncated negative binomial distribution with reproductive number R = 2.5 and dispersion parameter k = 0.1. In each experiment, we estimate averages using 10,000 samples. Double entries in the first column correspond to cases where the set of contacts is partitioned into a combination of pools of two different sizes.
(DOCX)

**S10 Table. Pool partitions corresponding to the points of Fig 3D, resulting by penalizing the false negative rate.** Here, under $s_e = 0.99$, $s_p = 0.99$, we **vary** $\lambda_1$ while we fix $\lambda_2 = 0$ and, for each resulting partition, we compute the average number of tests and false negative/positive rate. We set the number of contacts to N = 100 and sample the number of positive infections from a truncated negative binomial distribution with reproductive number R = 2.5 and dispersion parameter k = 0.1. In each experiment, we estimate averages using 10,000 samples. Double entries in the first column correspond to cases where the set of contacts is partitioned into a combination of pools of two different sizes.
(DOCX)

**S11 Table. Pool partitions corresponding to the points of Fig 3D, resulting by penalizing the false positive rate.** Here, under $s_e = 0.99$, $s_p = 0.99$, we **vary** $\lambda_2$ while we fix $\lambda_1 = 0$ and, for each resulting partition, we compute the average number of tests and false negative/positive rate. We set the number of contacts to N = 100 and sample the number of positive infections from a truncated negative binomial distribution with reproductive number R = 2.5 and dispersion parameter k = 0.1. In each experiment, we estimate averages using 10,000 samples. Double entries in the first column correspond to cases where the set of contacts is partitioned into a

combination of pools of two different sizes.
(DOCX)

**S1 Appendix. Derivations for Dorfman testing under overdispersion (Dorf-OD).**
(DOCX)

**S2 Appendix. Dynamic programming algorithm.**
(DOCX)

**S3 Appendix. Derivations for classic Dorfman's method (Dorf-Cl).**
(DOCX)

## Acknowledgments

We thank Vipul Bajaj for helpful conversations.

## Author Contributions

**Conceptualization:** Abir De, Manuel Gomez-Rodriguez.

**Data curation:** Stratis Tsirtsis.

**Formal analysis:** Stratis Tsirtsis, Abir De, Lars Lorch, Manuel Gomez-Rodriguez.

**Funding acquisition:** Manuel Gomez-Rodriguez.

**Investigation:** Stratis Tsirtsis.

**Methodology:** Stratis Tsirtsis, Abir De, Lars Lorch, Manuel Gomez-Rodriguez.

**Project administration:** Manuel Gomez-Rodriguez.

**Resources:** Manuel Gomez-Rodriguez.

**Software:** Stratis Tsirtsis.

**Supervision:** Manuel Gomez-Rodriguez.

**Validation:** Stratis Tsirtsis, Abir De, Lars Lorch, Manuel Gomez-Rodriguez.

**Visualization:** Stratis Tsirtsis.

**Writing – original draft:** Stratis Tsirtsis, Lars Lorch, Manuel Gomez-Rodriguez.

**Writing – review & editing:** Stratis Tsirtsis, Lars Lorch, Manuel Gomez-Rodriguez.

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
