## [Decision Letter · Decision Letter 0]

2 Dec 2021

Dear Mr. Tsirtsis,

Thank you very much for submitting your manuscript "Pooled Testing of Traced Contacts Under Superspreading Dynamics" for consideration at PLOS Computational Biology.

As with all papers reviewed by the journal, your manuscript was reviewed by members of the editorial board and by several independent reviewers. In light of the reviews (below this email), we would like to invite the resubmission of a significantly-revised version that takes into account the reviewers' comments.

We cannot make any decision about publication until we have seen the revised manuscript and your response to the reviewers' comments. Your revised manuscript is also likely to be sent to reviewers for further evaluation.

Sincerely,

Alex Perkins

Associate Editor

PLOS Computational Biology

Tom Britton

Deputy Editor

PLOS Computational Biology

Reviewer's Responses to Questions

**Comments to the Authors:**

Reviewer #1: The authors present an interesting new method to optimize test pooling in the context of contact tracing procedures. The work is well executed, both from a formal and experimental analysis point of view. The paper is well written and clear.

Some major remarks:

- My major concern with this work, is that the impact on the actual epidemic is not investigated. If I understand it correctly, temporal (e.g. viral load progression) and contact-related aspects of the epidemic are all aggregated in the clinical sensitivity/specificity (line 170). However, missing contacts, especially in an overdispersed epidemic context, could have implications on the overall attack rate. The authors mention (in the discussion), that this could be investigated through randomized control studies, yet I would argue that this could also be analysed by using an individual-based model to investigate the impact of your testing strategy? To this end, perhaps one of these individual-based models could be used [2,3,4]?

- One of your assumptions, early in the paper (line 100), where you assume that perfect tracing is possible, is quite a though one. While I understand that this assumption is necessary for your theoretical framework, I believe it would be good to acknowledge that this is not necessarily realistic and refer to the section where you empirically challenge this assumption (line 298).

Some minor remarks:

- Is there a reason why you use small r instead of R_0? This makes it an easier symbol to spot, and the symbol is quite commonly used in literature.

- Did you consider any additional population structures next to the contact tracing contacts (e.g., households), and how would this fit in your work?

- Adjacent to this, the related work section was quite complete, however I believe that the work on household-based pooling [1] could also be interesting to discuss.

- I found Figure 1 (b) a bit strange and hard to interpret at first sight. It looks strange with the negative percentages, could you perhaps show the test distributions for both methods instead next to each other?

- Why is Figure 2 ragged along the y-axis? Is this related to the experimental resolution you used to build this figure?

References:

[1] https://journals.plos.org/ploscompbiol/article?id=10.1371/journal.pcbi.1008688

[2] https://journals.plos.org/ploscompbiol/article?id=10.1371/journal.pcbi.1009149

[3] https://journals.plos.org/ploscompbiol/article?id=10.1371/journal.pcbi.1009146

[4] https://www.nature.com/articles/s41467-021-21747-7

Reviewer #2: Please see attached file.

Reviewer #3: The paper studies the problem of pooled testing in which the goal is identify an optimal pooling scheme that balances between testing costs and misclassification errors. The novel component is that the authors assume that arrivals are all contact traced to one person. The author model this using a truncated negative binomial distribution. The authors then show that their formulation can be solved via dynamic programming. Finally, the authors conduct an extensive numerical study in which they measure the benefits of their approach compared to classical Dorfman testing. Overall the paper is well-written and generally easy to follow. I do have some concerns about the modeling assumptions in addition to some clarifications which I list below. I recommend a major revision.

- Why use a truncated normal binomial and not some other mixture model? It would be interesting to observe the impact of that on the optimization outcome. The benefit of using a mixture model is that allows for more flexibility in choosing population variance which allows for the calibration of that variance using data.

- Page 4/33 line 142, what are the bounds of the summation in the objective function? The reason I ask this question is that it seems that the number of summands in the objective varies as you add more pools, which in itself is a decision in this problem.

- General comment on the case study. Since you are interested in minimizing a weighted sum of expected number of tests, false-negatives, and false-positives, then why don't you report the value of that objective function in your comparison of you approach with the Dorfman scheme?

- General question about the Dorfman scheme. It is not clear how this was implemented in the paper. How did you partition the set of N patients into pools to be tested using a Dorfman scheme? Maybe adding a paragraph about this in the experimental design section would help clarify this.

- Would it be possible to run your by only accounting for false negatives? It would be interesting to compare your approach to Dorfman testing if the only goal is to minimize negative misclassification errors.

**Have the authors made all data and (if applicable) computational code underlying the findings in their manuscript fully available?**

Reviewer #1: Yes

Reviewer #2: Yes

Reviewer #3: Yes

PLOS authors have the option to publish the peer review history of their article (what does this mean?). If published, this will include your full peer review and any attached files.

Reviewer #1: No

Reviewer #2: No

Reviewer #3: No
---

## [Decision Letter · Decision Letter 1]

28 Feb 2022

Dear Mr. Tsirtsis,

Thank you very much for submitting your manuscript "Pooled Testing of Traced Contacts Under Superspreading Dynamics" for consideration at PLOS Computational Biology. As with all papers reviewed by the journal, your manuscript was reviewed by members of the editorial board and by several independent reviewers. The reviewers appreciated the attention to an important topic. Based on the reviews, we are likely to accept this manuscript for publication, providing that you modify the manuscript according to the review recommendations.

Please address the one minor comment by Reviewer 1.

Sincerely,

Alex Perkins

Associate Editor

PLOS Computational Biology

Tom Britton

Deputy Editor

PLOS Computational Biology

[LINK]

Reviewer's Responses to Questions

**Comments to the Authors:**

Reviewer #1: I thank the authors for their answers and improvements of the manuscript.

One more remark: on my comment regarding the impact on the testing strategy on the epidemic, the authors state:

"That being said, we agree with the reviewer that it would be very interesting to investigate the

impact of different testing methods on the epidemic under a limited testing capacity using

individual-based models."

It would be good to indeed mention this in the discussion.

Reviewer #2: The authors addressed all my comments.

Reviewer #3: The authors addressed all of my comments and I have no further comments to add.

**Have the authors made all data and (if applicable) computational code underlying the findings in their manuscript fully available?**

Reviewer #1: Yes

Reviewer #2: Yes

Reviewer #3: Yes

PLOS authors have the option to publish the peer review history of their article (what does this mean?). If published, this will include your full peer review and any attached files.

Reviewer #1: No

Reviewer #2: No

Reviewer #3: No

Figure Files:

Data Requirements:

Reproducibility:

References:

---

## [Editor Report · Decision Letter 2]

10 Mar 2022

Dear Mr. Tsirtsis,

We are pleased to inform you that your manuscript 'Pooled Testing of Traced Contacts Under Superspreading Dynamics' has been provisionally accepted for publication in PLOS Computational Biology.

Best regards,

Alex Perkins

Associate Editor

PLOS Computational Biology

Tom Britton

Deputy Editor

PLOS Computational Biology

---

## [Editor Report · Acceptance letter]

24 Mar 2022

PCOMPBIOL-D-21-01826R2 

Pooled Testing of Traced Contacts Under Superspreading Dynamics

Dear Dr Tsirtsis,

I am pleased to inform you that your manuscript has been formally accepted for publication in PLOS Computational Biology. Your manuscript is now with our production department and you will be notified of the publication date in due course.

With kind regards,

Olena Szabo
